# Development of Simultaneous Analytical Method for Imidazolinone Herbicides from Livestock Products by UHPLC-MSMS

**DOI:** 10.3390/foods11121781

**Published:** 2022-06-16

**Authors:** Hyo-Min Heo, Hyeong-Wook Jo, Hee-Ra Chang, Joon-Kwan Moon

**Affiliations:** 1Hansalim Agro-Food Analysis Center, Hankyong National University Industry Academic Cooperation Foundation, Suwon 16500, Korea; hmheo@hknu.ac.kr (H.-M.H.); hyeongwook.jo@hknu.ac.kr (H.-W.J.); 2Department of Food & Pharmaceutical Engineering, Graduate School of Hoseo University, Asan 31499, Korea; hrchang@hoseo.edu; 3School of Plant Resources and Landscape Architecture, Hankyong National University, Anseong 17579, Korea

**Keywords:** livestock products, imidazolinone herbicides, simultaneous analytical method, LC-MS/MS

## Abstract

A simultaneous analytical method, which used LC/MSMS for imidazolinone herbicides from livestock products (egg, milk, beef, pork, and chicken) for monitoring, was developed with a QuEChERS preparation. A weighed sample (5 g) in a 50 mL conical tube was added to 0.1 M potassium phosphate dibasic solution (5 mL) and shaken for 10 min. After shaking, 0.5 mL of 6 N HCl and 5 mL of acetonitrile were added, and this solution was shaken for 10 min. Additionally, QuEChERS extraction salts (original method, 4 g MgSO_4_, 1 g NaCl) were added to the sample in a 50 mL conical tube. The mixture was strongly shaken for 1 min and centrifuged at 3000× *g* for 10 min. The acetonitrile layer was purified with dSPE (150 mg MgSO_4_, 25 mg C_18_) and was centrifuged at 13,000× *g* for 5 min. The supernatant was filtered with a membrane filter (pore size: 0.2 μm) before analysis. The ME (%, matrix effect) range for almost all analytes was −6.56 to 7.11%. MLOD (method limit of detection) and MLOQ (method limit of quantitative) values were calculated by the S/N ratio. MLOQs were 0.01 mg/kg. The linear correlation coefficients (R^2^) were >0.99 with the range of 0.5~25 μg/kg for all of the imidazolinone herbicides. The recoveries (of imidazolinone herbicides) were in the range of 76.1~110.6% (0.01 mg/kg level), 89.2~97.1% (0.1 mg/kg level), and 94.4~104.4% (0.5 mg/kg level). These are within the validation criteria (to recover 70–120% with RSD <20%). The method demonstrated the simple, rapid, high throughput screening and quantitative analysis of imidazolinone herbicide residues for monitoring in livestock products.

## 1. Introduction

Pesticides are indispensable agricultural materials that can protect crops from various pests, including insects, nematodes, mites, rodents, fungi, and weeds; these ensure high-quality agricultural products and stable income for the farmers [1,2,3,4]. These pesticides remain on crops through spraying or have a long half-life. In this case, they can also be transferred to livestock or humans through agricultural and food chains. Additionally, livestock are exposed to pesticides upon consumption of animal feedstuffs made from agricultural products such as cereal grains and straw [5,6,7]. Additionally, pollutants flow into various channels, such as livestock feed, soil, atmosphere, and pest infection prevention products [8]. In particular, as human living standards increase, the consumption of livestock foods also increases, so the safety of livestock products is of paramount importance. In 2019, the average daily intake of livestock foods was 33.1 g of eggs, 69.97 g of milk, 22.82 g of beef, 53.24 g of pork, and 30.62 g of chicken in Korea [9]. In 1989, 27 residue tolerance standards and 58 analysis methods were established in Korea for pesticide residue standards and inspection methods in livestock products (Ministry of Agriculture, Forestry and Fisheries). Since then, safety standards such as the ‘Livestock Product Residue Acceptance Standard’ for pesticides made from 19 ingredients, first introduced in 2004, have been continuously expanded and revised. Simultaneous multi-component analysis methods for livestock products should be developed in accordance with the continuous expansion and revision of residue limit standards. The analysis method was developed in accordance with the continuous expansion and revision of the residual tolerance standards for pesticide residues in livestock products.

Imidazolinone (imazapyr, imazamox, imazapic, imazethapyr, imazaquin, imazatabenz (free acid) and imazatabenz-methyl, Table 1) pesticides are herbicides used for the pre- and post-emergence control of annual and perennial grass, sedge, and broad-leaved weeds [10]. According to [10], the site of action of imidazolinone herbicides is known to inhibit branched-chain amino acid synthesis (acetolactate synthase or acetohydroxyacid synthase). These herbicides have characteristics such as a low toxicity to animals, birds, fish, and invertebrates, and they have been used in many cultivated crops due to their versatility, low toxicity, and low application rates [11]. Currently, MRLs for imidazolinone herbicides for livestock products are set as: imazapyr (pork; 0.05, chicken; 0.01, milk; 0.01, beef; 0.05, egg; 0.01 mg/kg), imazamox (pork; 0.05, chicken; 0.01, milk; 0.03, beef; 0.03, egg; 0.01 mg/kg), imazapic (pork; 0.1, chicken; 0.01, milk; 0.1, beef; 0.1, egg; 0.01 mg/kg), and imazethapyr (pork; 0.1, chicken; 0.1, milk; 0.1, beef; 0.1, egg; 0.1 mg/kg) in Japan [12]. Additionally, by CODEX [13], the MRLs are set as: imazapyr (meat; 0.05, chicken; 0.01, milk; 0.01, egg; 0.01 mg/kg), imazamox (meat; 0.01, chicken; 0.01, milk; 0.01, egg; 0.01 mg/kg), imazapic (meat; 0.1, chicken; 0.1, milk; 0.1, egg; 0.01 mg/kg), and imazethapyr (meat; 0.01, chicken; 0.01, milk; 0.01, egg; 0.01 mg/kg).

Analytical methods for the determination of imidazolinone herbicides in agriculture have been reported for most individual- or class-analytical methods [14,15,16,17,18,19,20]. Among domestic livestock products, no residual acceptance levels have been set. The current developed method of analyzing imidazolinone herbicides is to analyze samples such as soil and agricultural products, and extraction and purification methods using acids and bases were used to increase the efficiency of analysis according to the characteristics of each sample. Since it is difficult to apply the method directly to livestock products, it is necessary to develop an analysis method for the safety management of imidazolinone herbicides in livestock products. Therefore, through this study, we aim to develop a government-sponsored official analysis method for the determination of imidazolinone herbicide residues from livestock products to enable the expeditious monitoring and safety management. The samples were extracted and cleaned up using a modified QuEChERS preparation method for five kinds of livestock products (pigs, chickens, milk, cows, and eggs) prior to UHPLC-MSMS analysis. The method was validated according to the SANTE [21] and CODEX [22] guidelines.

## 2. Materials and Methods

### 2.1. Instrument

Sample analysis was performed in a UHPLC (Nexera X2, Shimadzu, Kyoto, Japan) coupled to a triple quadrupole mass spectrometer (8050, Shimadzu, Kyoto, Japan) system. The analytical column was operated using a Poroshell 120 SB-Aq (100 × 3.0 mm, 2.7 μm, Agilent, Santa Clara, CA, USA). The mobile phase employed a time-programmed gradient system using solvents A and B. Solvent A consisted of 0.1% formic acid and 5 mM ammonium formate in water, whereas solvent B was 0.1% formic acid and 5 mM ammonium formate in methanol. Gradient elution was initiated with 70% A for 2.0 min, and solvent B was increased to 50% after 3.0 min and 60% within 5.0 min. Solvent B was further gradually increased to 80% within 7.0 min, and 95% within 10.0 min. Then, it was kept constant for 1.0 min. Finally, B was linearly decreased to 30% over 11.1 min and equilibrated for 3.9 min. The total analytical time was 15.0 min, and the injection volume was 10 uL. The flow rate was 0.3 mL/min. The conditions for mass spectrometry were set as follows: for the MRM (multi reaction monitoring) mode the interface temperature was 300 °C; heat block temperature was 400 °C; DL temperature was 220 °C; nebulizing gas flow was 3 L/min; heating gas flow was 10 L/min; and drying gas flow was 10 L/min. In this study, MRM analyses were carried out in positive mode for regular detection. The MRM parameters, precursor ions, product ions, and collision energy are shown in Table 2 and Figure 1.

### 2.2. Reagents and Materials

Methanol (MeOH) and acetonitrile (MeCN) were obtained from Honeywell (Muskegon, MI, USA) at HPLC grade. Ammonium formate (≥99.0%) and potassium phosphate dibasic (≥98.0%, ACS reagent) were purchased from Sigma-Aldrich (St. Louis, MO, USA). Formic acid was obtained from Merck (Darmstadt, Germany) with an available purity of 99.0%. Hydrochloric acid (HCl, 36.0~38.0%, electronic grade) was obtained from Duksan pure Chemicals (Ansan, Korea). Sodium chloride, anhydrous Na_2_SO_4_, trisodium citrate, disodium citrate, anhydrous MgSO_4_, primary secondary amine (PSA), octadecyl silane (C_18_), and graphitized carbon black (GCB) were purchased from Agilent (Santa Clara, CA, USA).

### 2.3. Preparation of Standard Solution

Individual standards were purchased from Dr. Ehrenstorfer GmbH (Augsburg, Germany). Stock solutions of 1000 mg/L were prepared in MeCN according to their solubility. The working solution mixtures were prepared for the calibration curve by diluting the stock solutions with MeCN. The solutions were then kept in the dark at −20 °C in an amber glass vial before use.

### 2.4. Sample Preparation

A modified QuEChERS sample preparation method was used. The extraction step was evaluated with different solvents (using 6N HCl solution and MeCN for the second extraction after using 0.1 M ammonium acetate solution and 0.1 M potassium phosphate solution for the first extraction). After extraction, the samples were treated with salt 1 (4 g anhydrous magnesium sulfate and 1 g sodium chloride), salt 2 (4 g anhydrous magnesium sulfate, 1 g sodium chloride, 1 g trisodium citrate dihydrate and 0.5 g disodium hydrogen citrate sesquihydrate), or salt 3 (6 g anhydrous magnesium sulfate and 1.5 g sodium acetate), and shaken for 1 min. The extracted samples were then centrifuged at 4000 rpm for 10 min. Then, the clean-up step was evaluated using different sorbents, including MgSO_4_, PSA, C_18_, and GCB, and each of them was tested through pairing with MgSO_4_. The final steps were performed as follows: 5 g livestock product samples were weighed into 50 mL conical tubes, spiked with standard solution, and allowed to stand for 30 min at room temperature. During the extraction process, 5 mL of 0.1 M potassium phosphate solution was added into the samples and mixed on a mechanical wrist shaker for 10 min (first extraction). Then, 0.5 mL of 6N HCl solution and 10 mL MeCN were added into the samples and mixed on a mechanical wrist shaker for 10 min (second extraction). To stratify the organic phase and water, 4 g anhydrous magnesium sulfate and 1 g sodium chloride were added to each tube. After 1 min of shaking and centrifuging (4000 rpm for 10 min), 1 mL of the upper layer (organic phase) was transferred to 1.5 mL centrifuge tubes containing 150 mg MgSO_4_ and 25 mg C_18_, which were shaken for 1 min. The tubes were centrifuged at 12,000 rpm for 5 min at 4 °C. One hundred µL aliquots of supernatant were transferred to a microtube and mixed with 800 µL buffer solution (100 mM ammonium formate in water at pH 4~4.5, adjusted with formic acid) and 100 µL MeCN. Additionally, the final mixtures were filtered with a 0.2 µm membrane (PTFE) before being injected into the UHPLC-MS/MS.

### 2.5. Validation of the Method

For the validation of the method, blank livestock samples were selected, including egg, milk, beef, pork, and chicken, and validation parameters were evaluated, including calibration curve linear range (linearity), accuracy, repeatability, precision, method limits of detection (MLOD), and quantification (MLOQ). The linearity was evaluated using matrix matched calibration curves at 0.25, 0.5, 5, 10, 20, and 50 µg/L prepared in MeCN, buffer solution, and matrix blank extraction. The matrix effect was estimated by comparing the slopes of the curves in matrix blank extraction and solvent (MeCN). The difference in the slopes of the matrix extraction and solvent curves were divided by the slope of the solvent curve and expressed as % of matrix effect. Accuracy was evaluated through recovery testing, spiking the blank samples (beef, pork, chicken, egg, and milk) at 10, 100, and 500 µg/kg, with 5 replicates performed for each spiked level to determine the precision of the method. The repeatability and precision of the method were also evaluated through relative standard deviation (RSD%) below 20% of five replicates. All the analytes were determined in a laboratory accredited to the SANTE/12682/2019 and CODEX guidelines, which followed these quality criteria.

## 3. Results

### 3.1. Optimization of the QuEChERS Procedure

Different procedures based on the QuEChERS method have been modified as follows.

#### 3.1.1. Optimization of the Extraction Solution and Salts

pH may play an essential role in extracting imidazolinone herbicides during the extraction process because they contain a carboxyl group and amine group. Since hydrogen bound to a carboxyl group or an amine group is disassociated and does not transfer to an organic solvent, or is adsorbed to an interfering substance, its extraction efficiency is lowered. Therefore, the effect of pH on imidazolinone herbicides recovery has been investigated in many studies [20,23,24,25,26,27]. For the efficiency of the extraction solvent according to pH, the extraction solvent was selected using an alkaline and an acidic material. As shown in Table 3, recovery rates in 0.1 M potassium phosphate solution (pH 9.42) were higher than those in 0.1 M ammonium acetate solution (pH 7.11). As a result, it was found that the higher the pH of the imidazolinone herbicides, the higher the extraction efficiency of the aqueous layer. Additionally, by adding 0.5 mL of 6 N HCl (pH 2.5) to increase the log P_ow_, a high recovery rate was obtained: from 92.0 to 100.9%. For the addition of salt to the distribution of the aqueous solution layer and the organic solvent layer, the salt (no buffer is added, 4 g MgSO_4_ and 1 g NaCl) that has the least influence on the pH during extraction was selected.

#### 3.1.2. Optimization of the Purification Adsorbent

A d-SPE (Dispersive Solid-Phase Extraction) clean-up step was evaluated using d-SPE composed of existing commercially available adsorbents. The initial clean-up step was tested with eight sorbents (1; 50 mg PSA, 50 mg C_18_, 50 mg GCB, 2; 25 mg PSA, 50 mg GCB, 3; 50 mg PSA, 4; 50 mg PSA, 50 mg GCB, 5; 25 mg PSA, 25 mg GCB, 6; 50 mg PSA, 50 mg C_18_, 7; 25 mg C_18_, 8; 25 mg PSA, 25 mg C_18_) and each of them was mixed with 150 mg MgSO_4_. The recovery rates were less than 10% in the eight sorbents containing PSA excluding imazamethabenz-methyl. C_18_ is a reverse-phase adsorption material that removes non-polar interfering substances, such as lipids, cholesterol, and lipophilic compounds. PSA is a weak anion exchange adsorption material that can adsorb polar molecules and effectively remove co-extracted components from the matrix, such as sugars and organic acids. Because of these characteristics, the imidazolinone herbicides are weakly acidic, and when an adsorbent containing PSA is used, the purification efficiency seems to be significantly lowered (Figure 2).

According to the guidelines [21,22], the acceptable recovery rate is 70–120%, with an RSD less than or equal to 20% for multi-residue methods. Of the eight adsorption combinations, only 150 mg MgSO_4_ and 25 mg C_18_ satisfied this acceptable recovery rate. Therefore, 150 mg MgSO_4_ and 25 mg C_18_ were used as d-SPE clean-up livestock products in this study.

### 3.2. Matrix Effect

Co-eluting interfering substances, such as fats, lipids, and proteins in livestock products, interfere with the ionization of pesticides with the suppression or the enhancement of the response. Matrix effects were calculated with Equation (1) as follows:(1)$$\mathrm{Matix}\;\mathrm{effect}\left(\mathrm{ME},\;\%\right)=\left(\frac{Slope\;of\;the\;matrix\;standard\;curve}{Slope\;of\;the\;solvent\;standard\;curve}-1\right)\times 100$$

ME can be classified into three ranges based on the results of the calculated data (strong matrix effect: lMEl ≥ 50; medium matrix effect: 20 < lMEl < 50; and small matrix effect: lMEl ≤ 20) [28]. As shown in Table 4, all of the imidazolinone herbicides had a small matrix effect in livestock products.

### 3.3. Method Validation

The linearity, MLOD, MLOQ, accuracy, and precision were determined to evaluate the performance of the modified QuEChERS prepared method. The linearity was collected in the 0.25–50 µg/L concentration range. As presented in Table 5, the coefficients of determination (R^2^) were higher than 0.99 for the imidazolinone herbicides.

Accuracy was evaluated at spiked concentrations of 10, 100, and 500 µg/kg in livestock products. The average recoveries of seven imidazolinone herbicides ranged from 76.1% to 110.6%. The lowest accuracy value was relative to imazethapyr (76.1%) in 10 µg/kg. In the analysis of pesticide residues using mass spectrometry, the recovery rate may exceed 100% depending on the change in sensitivity during the ionization process due to interfering substances in the matrices, and the recovery rate may be lower than 100% due to adsorption with interfering substances during extraction or purification. For this reason, the analytical method validation guidelines specify the recovery range differently for each concentration (Table 6). Thus, the method’s precision can be considered appropriate (SANTE/12682/2019). For seven imidazolinone herbicides, the RSD values ranged from 0.7% to 8.4% under laboratory conditions in all recovery experiments, an indication that the method’s precision was acceptable. Therefore, it can be concluded that the modified QuEChERS prepared method is quick and accurate in determining the residues of the monitored pesticides in livestock products. The experimental results of the method performance evaluation, including recovery values (Rec, %), standard deviation, and RSD (%), are shown in Table 7.

## 4. Discussion

Imidazolinone herbicides are weakly acidic compounds containing an imidazoline ring, capable of decomposing intramolecular acids (H^+^) into polar compounds. Therefore, the electro-spray ionization positive mode (ESI) was operated using ionizable features. In addition, HPLC (high-performance liquid chromatography) was used because it was expected that GC (gas chromatography) analysis would be difficult because it was a nonvolatile material with a low vapor pressure. It has a characteristic that depends on pH. In other studies, when extracting imidazolinone herbicides, a buffer capable of adjusting the pH was added, and then extraction was carried out [23,24,25,26,27]. Secondary extraction was performed when analyzing imidazolinone herbicides in soil, and the effect of extraction using acetate with a concentration of pH 5.5 was used. In addition, PSA was used to increase the purification effect [17]. However, in this study, after the first extraction using a solution containing potassium of pH 9.42, a second extraction method was performed to lower the pH using a strong acid (6N HCl, pH 2.5), and the fatty interfering material was removed using C_18_. The ratio of the final solvent was efficient at 2/8 of MeCN/buffer solution (*v*/*v*). The ratio of the extraction solvent and the buffer solvent was set using polar properties. Additionally, it was observed that the recovery rate was high for each standard solution when the pH was dropped and extracted as MeCN by adding acid after the first extraction with a solvent of pH 8 or higher. Partition efficiency was greatest in salt composed of 4 g MgSO_4_ and 1 g sodium chloride to minimize the effect of pH. Additionally, the purification efficiency was highest in a composition of 150 mg MgSO_4_ and 25 mg C_18_. Previously, studies on imidazolinone herbicides have mostly been conducted on soil, and most of the pretreatment methods used were SPE or d-SPE after extraction by adjusting the pH [18,19,20,23,24,25,26,27].

## 5. Conclusions

A validation for a simultaneous analytical method based on modified QuEChERS and LC-MS/MS was established to rapidly analyze multi-residue pesticides in livestock products. The modified QuEChERS sample preparation method uses an original salting agent and then a purification treatment with 150 mg MgSO_4_ and 25 mg C_18_ added, which effectively removes interference and reduces the matrix effect of imidazolinone herbicides in livestock products. Overall, seven pesticides passed the validation with satisfactory recoveries (70–120%) and an RSD of ≤20%. These results show that the method is effective, easy, quick, and reliable for the routine monitoring of pesticide residues in livestock products.

## Figures and Tables

**Figure 1 foods-11-01781-f001:**
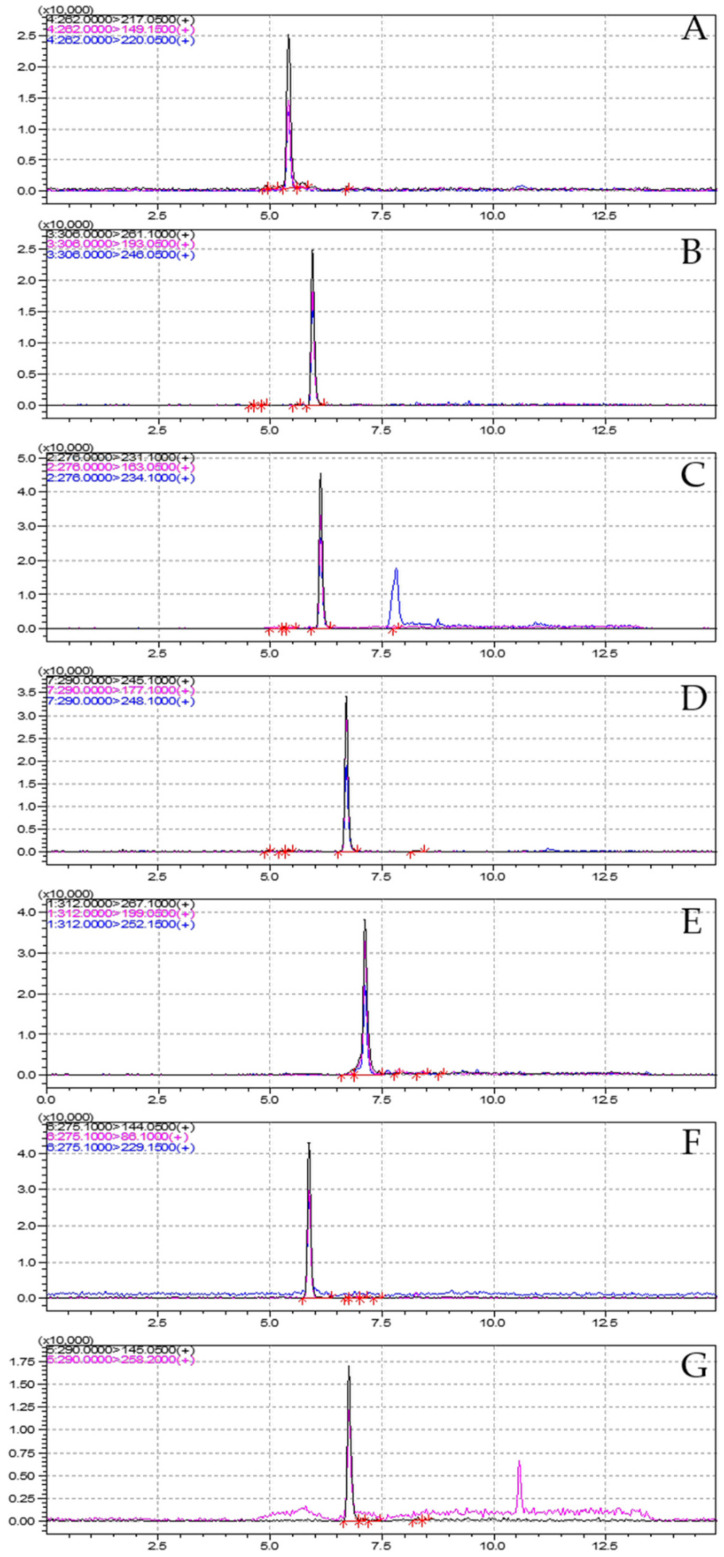
Chromatograms (10 µg/L) of seven imidazolinone herbicides ((**A**): Imazapyr, (**B**): Imazamox, (**C**): Imazapic, (**D**): Imazethapyr, (**E**): Imazaquin, (**F**): Imazamethabenz (free acid), (**G**): Imazamethabenz-methyl).

**Figure 2 foods-11-01781-f002:**
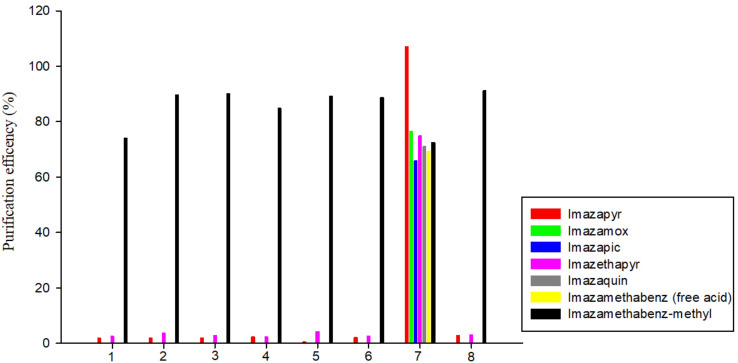
Efficiency of eight purification adsorbents (1; 50 mg PSA, 50 mg C_18_, 50 mg GCB, 2; 25 mg PSA, 50 mg GCB, 3; 50 mg PSA, 4; 50 mg PSA, 50 mg GCB, 5; 25 mg PSA, 25 mg GCB, 6; 50 mg PSA, 50 mg C_18_, 7; 25 mg C_18_, 8; 25 mg PSA, 25 mg C_18_).

**Table 1 foods-11-01781-t001:** Physico-chemical properties of imidazolinone herbicides.

Compound	Cas No.	Molecular Weight (g/mol)	*p*Ka	Log Pow	Vapor Pressure (mPa)	Solubility (g/L)	MRL (mg/kg)		Structure
							CODEX	Japan	
Imazapyr	81334-34-1	261.3	3.6 11.0 1.9	0.11	<0.013	Acetone 33.9 Methanol 1.05 Water 11.3	0.05 (meat) 0.01 (chicken) 0.01 (milk) 0.01 (egg)	0.05 (pork, beef) 0.01 (chicken) 0.01 (milk) 0.01 (egg)	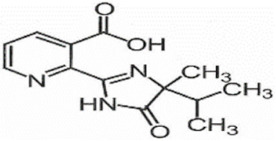
Imazamox	114311-32-9	305.3	2.3 10.8 3.3	−0.9 (pH 7) −0.3 (pH 4)	6.3 × 10^−8^	Acetone 29.3 Methanol 67 Water 4.16	0.01(meat) 0.01 (chicken) 0.01 (milk) 0.01 (egg)	0.05 (pork) 0.03 (beef) 0.01 (chicken) 0.03 (milk) 0.01 (egg)	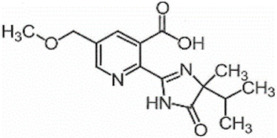
Imazapic	104098-48-8	275.3	11.1 3.6 2.0	0.393	<0.01	Acetone 18.9 Water 2.15	0.1 (meat) 0.1 (chicken) 0.1 (milk) 0.01 (egg)	0.1 (pork) 0.1 (beef) 0.1 (chicken) 0.1 (milk) 0.01 (egg)	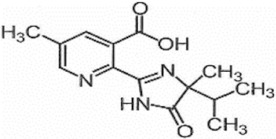
Imazethapyr	81335-77-5	289.3	2.1 3.9	1.2 (pH 9) 1.49 (pH 7) 1.04 (pH 5)	<0.013	Acetone 48.2 Methanol 105 Water 1.4	0.05 (meat) 0.01 (chicken) 0.01 (milk) 0.01 (egg)	0.1 (pork) 0.1 (beef) 0.1 (chicken) 0.1 (milk) 0.1 (egg)	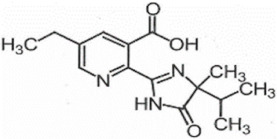
Imazaquin	81335-37-7	311.3	3.45 11.03	−1.32 (pH 10) −1.09 (pH7) 0.833 (pH 4)	2 × 10^−9^	Acetone 3.69 Methanol 5.77 Water 102	Not set	Not set	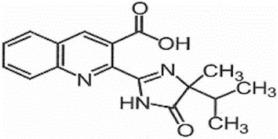
Imazamethabenz (free acid)	89318-82-1	274.3	-	1.9	<0.013	Water 0.074	Not set	Not set	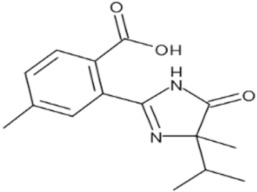
Imazamethabenz -methyl	81405-85-8	288.3	3.1	1.9	0.0021	Acetone 180 Methanol 244 Water 2.2	Not set	Not set	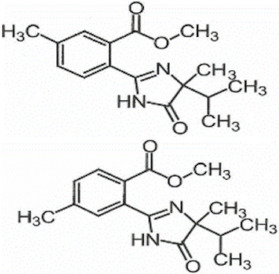

**Table 2 foods-11-01781-t002:** Optimized MRM parameters for the determination of imidazolinone herbicides.

Compound Name	RT (min)	Ionization	Precursor Ion > Product Ion (CE, eV)			
			Quantifier Ion	CE	Qualifier Ion	CE
Imazapyr	5.419	[M+H]^+^	262.0 > 217.05	−20	262.0 > 220.05	−18
Imazamox	5.959	[M+H]^+^	306.0 > 261.10	−21	306.0 > 246.05	−25
Imazapic	6.136	[M+H]^+^	276.0 > 231.10	−21	276.0 > 163.05	−26
Imazethapyr	6.720	[M+H]^+^	290.0 > 245.10	−21	290.0 > 177.10	−30
Imazaquin	7.141	[M+H]^+^	312.0 > 267.10	−22	312.0 > 199.05	−29
Imazamethabenz (free acid)	5.882	[M+H]^+^	275.1 > 144.05	−36	275.1 > 229.15	−20
Imazamethabenz -methyl	6.782	[M+H]^+^	290.0 > 230.15	−20	290.0 > 145.05	−36

**Table 3 foods-11-01781-t003:** Efficiency of extraction solvent towards imidazolinone herbicides.

Compound Name	First Extraction		Add Acid Material
	0.1 M Ammonium Acetate	0.1 M Potassium Phosphate	6N HCl
Imazapyr	0.7	85.0	96.2
Imazamox	3.1	88.2	97.1
Imazapic	2.1	93.4	98.4
Imazethapyr	4.1	92.8	94.9
Imazaquin	5.0	93.1	100.9
Imazamethabenz (free acid)	5.0	89.5	92.0
Imazamethabenz -methyl	6.3	90.5	98.7

**Table 4 foods-11-01781-t004:** Matrix effects for imidazolinone herbicides to livestock products.

Compound Name	Matrix Effect (% ME)				
	Egg	Milk	Beef	Pork	Chicken
Imazapyr	1.99	−4.70	−0.77	7.11	−0.27
Imazamox	3.13	−3.41	1.39	1.67	1.73
Imazapic	−1.76	−3.72	−1.03	1.63	−0.63
Imazethapyr	−4.72	−5.09	−1.18	0.52	−2.00
Imazaquin	−0.90	−2.12	−0.92	0.53	0.66
Imazamethabenz (free acid)	−6.56	−3.70	−3.12	3.01	−6.02
Imazamethabenz-methyl	−2.71	−3.38	0.42	5.70	0.10

**Table 5 foods-11-01781-t005:** MLOD, MLOQ, and linearity for imidazolinone herbicides to livestock products.

Compound Name	Limit of Detection (mg/kg)	Limit of Quantification (mg/kg)	Linearity (R^2^)				
			Egg	Milk	Beef	Pork	Chicken
Imazapyr	0.0005	0.01	0.9999	0.9999	0.9998	0.9999	0.9999
Imazamox	0.0005	0.01	0.9999	0.9999	0.9998	0.9999	0.9999
Imazapic	0.0005	0.01	0.9999	0.9998	0.9999	0.9999	0.9998
Imazethapyr	0.0005	0.01	0.9998	0.9999	0.9999	0.9999	0.9999
Imazaquin	0.0005	0.01	0.9999	0.9999	0.9998	0.9998	0.9999
Imazamethabenz (free acid)	0.0005	0.01	0.9999	0.9999	0.9999	0.9999	0.9999
Imazamethabenz-methyl	0.0005	0.01	0.9996	0.9996	0.9999	0.9999	0.9998

**Table 6 foods-11-01781-t006:** In-laboratory method validation criteria for analysis of pesticide residues.

Concentration (mg/kg)	Repeatability	Trueness (Range of Mean % Recovery)
x≤0.001	35	50–120
0.001<x≤0.01	30	60–120
0.01<x≤0.1	20	70–120
0.1<x≤1	15	70–110
1<x	10	70–110

**Table 7 foods-11-01781-t007:** Recovery of imidazolinone herbicides obtained from QuEChERS sample preparation (*n* = 5).

**Compound Name**	Fortification Level (μg/kg)	Recovery (%)														
		Egg			Milk			Beef			Pork			Chicken		
		Aver ^1^	STDEV ^2^	% RSD ^3^	Aver ^1^	STDEV ^2^	% RSD ^3^	Aver ^1^	STDEV ^2^	% RSD ^3^	Aver ^1^	STDEV ^2^	% RSD ^3^	Aver ^1^	STDEV ^2^	% RSD ^3^
Imazapyr	10	82.7	2.4	2.9	106.2	3.3	3.1	108.0	5.7	5.2	93.3	7.0	7.5	102.4	5.5	5.3
	100	92.3	1.1	1.2	94.5	1.8	1.9	94.7	1.5	1.6	89.2	1.4	1.6	91.3	1.4	1.6
	500	95.4	1.4	1.4	98.5	1.4	1.4	103.8	2.2	2.2	98.5	2.7	2.8	95.8	1.3	1.4
Imazamox	10	89.2	4.9	5.5	105.2	5.6	5.3	106.6	3.3	3.1	107.5	7.0	6.5	104.3	5.0	4.8
	100	93.5	1.2	1.3	94.0	2.0	2.1	94.9	1.9	2.0	93.8	2.8	3.0	91.5	0.9	1.0
	500	96.9	2.7	2.8	99.4	1.4	1.4	104.4	1.7	1.7	100.6	2.8	2.8	96.8	2.2	2.3
Imazapic	10	91.5	2.1	2.3	87.6	4.6	5.3	95.5	4.8	4.8	99.2	7.5	7.6	91.1	1.6	1.7
	100	95.5	0.9	0.9	93.8	1.8	1.9	91.3	1.7	1.8	90.8	0.6	0.7	90.8	1.5	1.7
	500	97.6	0.9	1.0	96.4	1.3	1.4	97.6	1.0	1.0	100.0	3.2	3.2	94.4	1.7	1.7
Imazethapyr	10	76.1	1.2	1.5	87.3	2.0	2.3	106.3	6.7	6.3	96.8	4.1	4.3	95.8	8.1	8.4
	100	93.9	1.7	1.8	94.2	1.3	1.4	94.3	2	2.1	94.7	2.4	2.5	92.5	1.5	1.6
	500	94.6	1.8	1.9	96.4	1.4	1.5	99.0	1.0	1.0	102.5	0.9	0.9	94.8	1.7	1.7
Imazaquin	10	88.3	4.5	5.1	92.1	4.6	5.0	101.7	4.4	4.3	99.4	4.0	4.0	86.3	2.6	3.0
	100	94.2	1.2	1.5	96.8	1.5	1.5	97.1	2.6	2.7	96.7	2.9	3.0	90.8	1.2	1.3
	500	95.3	2.7	2.8	99.3	1.0	1.0	99.5	3.3	3.3	101.0	1.4	1.3	94.4	1.4	1.5
Imazamethabenz (free acid)	10	103.3	4.0	3.9	104.4	2.4	2.3	107.5	1.5	1.4	110.6	5.3	4.8	106.2	2.3	2.2
	100	95.0	1.6	1.7	96.8	1.5	1.5	92	0.7	0.8	92.1	1.3	1.4	90.9	1.2	1.3
	500	99.8	2.7	2.8	99.3	1.0	1.0	102.1	2.1	2.1	100.2	0.9	0.9	97.2	2.1	2.1
Imazamethabenz -methyl	10	89.9	2.3	2.6	93.4	2.0	2.2	102.1	4.3	4.3	87.6	3.9	4.4	84.1	4.1	4.9
	100	96.5	0.7	0.7	94.5	1.4	1.5	95.5	1.4	1.4	93.9	1.9	2.0	91.4	1.5	1.6
	500	97.6	2.3	2.3	98.4	2.2	2.2	97.7	1.0	1.0	95.7	1.6	1.7	94.6	1.2	1.3

^1^ Aver: average, ^2^ STDEV: standard deviation, ^3^ % RSD: relative standard deviation.

## Data Availability

Data is contained within the article.

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
