# Peer review of "Development of Simultaneous Analytical Method for Imidazolinone Herbicides from Livestock Products by UHPLC-MSMS"

_foods, 2022, doi:10.3390/foods11121781_

Round 1
Reviewer 1 Report
The paper entitled ‘Development of Simultaneous Analytical Method for Imidazolinone Herbicides from Livestock Products by UHPLC- MSMS’ reported quantitative analysis of imidazolinone herbicides residues for monitoring in livestock products. I have no specific scientific concern but it is mandatory to correct the manuscript in some points:
1. In Table 1, a CAS NO. is still need to be supplemented. If it's available, MRLs for these imidazolinone herbicides should be also list in the table. This would make this work way more useful and readable.
2. I also suggest redraw Figure 2 to get the aesthetics.
3. In 3.2, please added calculation formula of matrix effect.
4. The format of references should be revised.
Author Response
- edited Table 1
- redrawed Figure 2
- Added calculation formula of matrix effect
- Revised the format of references

Reviewer 2 Report
Foods
foods-1780056
Development of Simultaneous Analytical Method for Imidazolinone Herbicides from Livestock Products by UHPLC-MSMS
Dear Editor,
The article deals with the method development for imidazolinone herbicides from livestock products. The topic is good. The manuscript has been well designed and written. The discussion and interpretation sections are perfect. It can be accepted and published after necessary minor revisions are done. My questions and comments are below;
- Line 17: check the word “t ube”
- Lines 46 and 47: For where?
- Line 179: Can you explain this sentence a little more?
- Line 237: What are the possible reasons for the recovery rate to exceed 100%?
Author Response
- 17 line checked: 't ube' word
- added country
- line 181, explain more
- The recovery rate may exceed 100% due to the matrix or ionization process and the recovery rate may be lower than 100% due to adsorption with interfering substances during extraction or purification
